# The November 2019 Seismic Sequence in Albania: Geodetic Constraints and Fault Interaction

**Alessandro Caporali [1,2,\*] , Mario Floris [1] , Xue Chen [1] , Bilbil Nurce [3] , Mauro Bertocco [2] and Joaquin Zurutuza [2]**

1   Department of Geosciences, University of Padova, 35131 Padova, Italy; mario.floris@unipd.it (M.F.); chenxue@cugb.edu.cn (X.C.)
2   Center for Space Activities CISAS 'G. Colombo', University of Padova, 35131 Padova, Italy; mauro.bertocco@unipd.it (M.B.); jzurutuza@gmail.com (J.Z.)
3   Department of Geodesy, Universiteti Politeknik i Tiranës, Tirana 1001, Albania; billnurce@gmail.com
\*   Correspondence: alessandro.caporali@unipd.it; Tel.: +39-049-827-9122

**Abstract:** The seismic sequence of November 2019 in Albania culminating with the Mw = 6.4 event of 26 November 2019 was examined from the geodetic (InSAR and GNSS), structural, and historical viewpoints, with some ideas on possible areas of greater hazard. We present accurate estimates of the coseismic displacements using permanent GNSS stations active before and after the sequence, as well as SAR interferograms with Sentinel-1 in ascending and descending mode. When compared with the displacements predicted by a dislocation model on an elastic half space using the moment tensor information of a reverse fault mechanism, the InSAR and GNSS data fit at the mm level provided the hypocentral depth is set to 8 ± 2 km. Next, we examined the elastic stress generated by the Mw = 7.2 Montenegro earthquake of 1979, with the Albania 2019 event as receiver fault, to conclude that the Coulomb stress transfer, at least for the elastic component, was too small to have influenced the 2019 Albania event. A somewhat different picture emerges from the combined elastic deformation resulting after the two (1979 and 2019) events: we investigated the fault geometries where the Coulomb stress is maximized and concluded that the geometry with highest induced Coulomb stress, of the order of ca. 2–3 bar (0.2–0.3 MPa), is that of a vertical, dextral strike slip fault, striking SW to NE. This optimal receiver fault is located between the faults activated in 1979 and 2019, and very closely resembles the Lezhe fault, which marks the transition between the Dinarides and the Albanides.

**Keywords:** InSAR; GNSS; coseismic displacements; Coulomb stress transfer; seismic hazard

## 1. Introduction and Tectonic Setting

The Mw = 6.4 earthquake with epicenter near Durazzo in Albania (Lat = 41.46°N; Long = 19.58°E) at an estimated depth of 26 km (http://geofon.gfz-potsdam.de/eqinfo/event.php?id=gfz2019xdig) took place on 26 November 2019 at 2:54 UTC. The event was preceded by weaker shocks (max Mw = 5.6 on 21 September 2019) and followed by a swarm of aftershocks within some 10s km, with the largest event being of Mw = 5.5 on the same day of 26 November 2019 at 06:08:24 UTC. The epicenters of the aftershocks appear to migrate in the north direction (https://www.emsc-csem.org/Earthquake/262/M6-4-ADRIATIC-SEA-on-November-26th-2019-at-02-54-UTC#aftershocks). The preliminary fault plane solutions indicate a reverse fault with a high dip fault plane (strike = 151°, dip= 72°) with SW vergence and a lower dip fault plane (strike = 335°, dip = 18°) with NE vergence (Figure 1 and Table 1).

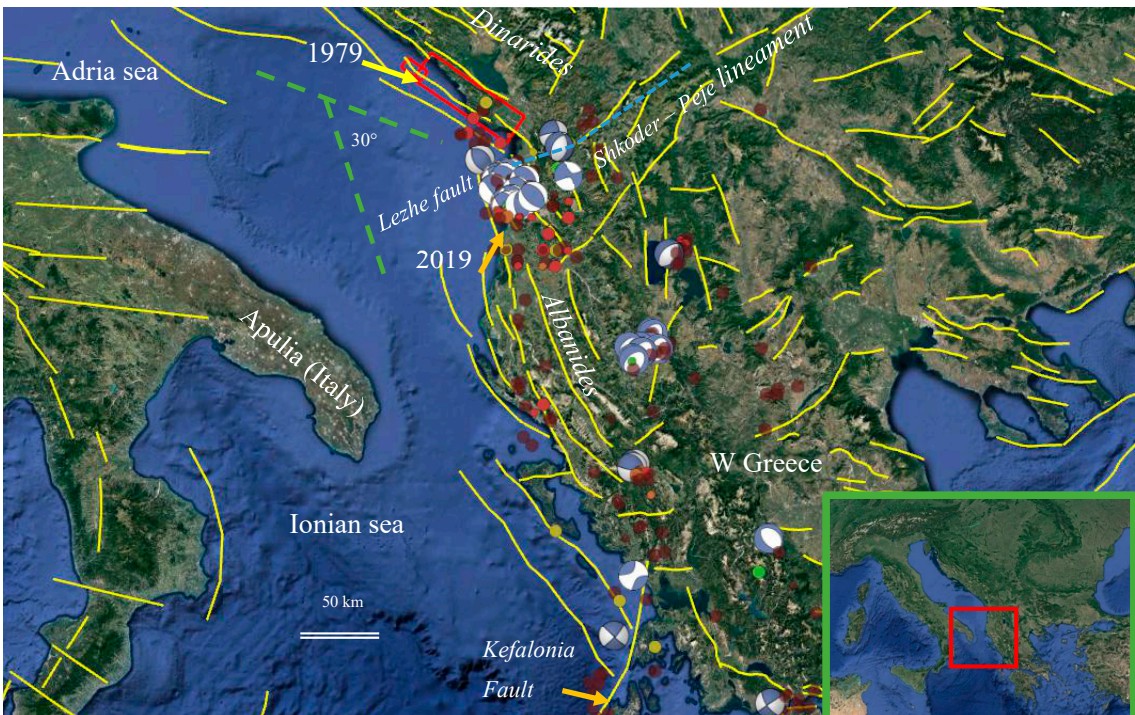

**Figure 1.** Location of the 2019 main event of Mw = 6.4 and seismicity of known and unknown (colored dots) fault plane solution since 2003 (Source: https://geofon.gfz-potsdam.de/eqinfo/form.php). The yellow lines indicate tectonic lineaments according to the DISS-Share Database [1,2]. The presumable fault plane of the 1979 Montenegro earthquake of Mw = 7.2 is also shown (red rectangles).

**Table 1.** Fault plane solutions of the events of magnitude greater than 5, published by GFZ (http://geofon.gfz-potsdam.de/eqinfo/). The hypocentral depth of the main event is debated (see the next sections).

| Date (UTC) | Lat (deg) | Long (deg) | Depth (km) | $M_W$ | Strike1 (deg) | Dip1 (deg) | Rake1 (deg) | Strike2 (deg) | Dip2 (deg) | Rake2 (deg) |
|---|---|---|---|---|---|---|---|---|---|---|
| 26 November 2019 2:54 | 41.46 | 19.58 | 26 | 6.4 | 151 | 72 | 89 | 335 | 18 | 98 |
| 26 November 2019 6:08 | 41.54 | 19.42 | 26 | 5.5 | 139 | 65 | 85 | 332 | 22 | 102 |
| 27 November 2019 14:45 | 41.54 | 19.42 | 26 | 5.3 | 155 | 63 | 89 | 337 | 27 | 91 |

The Composite Seismogenic Source is identified in the Database of Individual Seismogenic Sources (DISS) of INGV as ALCS (DISS Working Group, 2018) (http://diss.rm.ingv.it/dissnet/CadmoDriver?_action_do_single=1&_state=find&_token=NULLNULLNULLNULL&_tabber=1&_page=pSASources_d&IDSource=ALCS002). This structure belongs to the External Albanides, a compressive belt that is bordered to the N by the Dinarides, a ca. 700 km long belt overriding the Adria microplate. The two belts differ in strike by some 30°. The change in strike roughly corresponds to the Shkoder–Peje lineament (Figure 1), a major SW to NE striking structure [3] caught in between two thrust zones. Near to its SW termination, the Lezhe fault was described as a strike slip fault by Aliaj et al. [4]. Additional right lateral strike slip faults decouple the NE motion associated to the counterclockwise rotation of the Italian peninsula from the counterclockwise rotation of the Balkan–Hellenides towards SW [5]. The most notable example is the Kefalonia–Lefkada Transform Fault, which is characterized by repeated seismicity [6] (Figure 1). While the Dinarides/Albanian front moves SW, relative to a stable Europe, essentially as a rigid block [7], there is a clear indication that the Apulia region of the Italian peninsula is aseismic and moves NE converging towards the Dinarides. One expects a remarkable right lateral shear region to be accommodated in the mountain chains striking Albania and Western Greece NW to SE, just north of the complicated and intense deformation pattern affecting the West Hellenic arc [8–10]. The regional scale dextral shear field is well described by

the pattern of velocities of GNSS stations (Figure 2). In conclusion, the thrusts in the Lower Dinarides, Albanides, and Western Greece need to be interleaved with dextral strike slip faults to accommodate the opposite (i.e., in the NE direction) motion of the Italian peninsula.

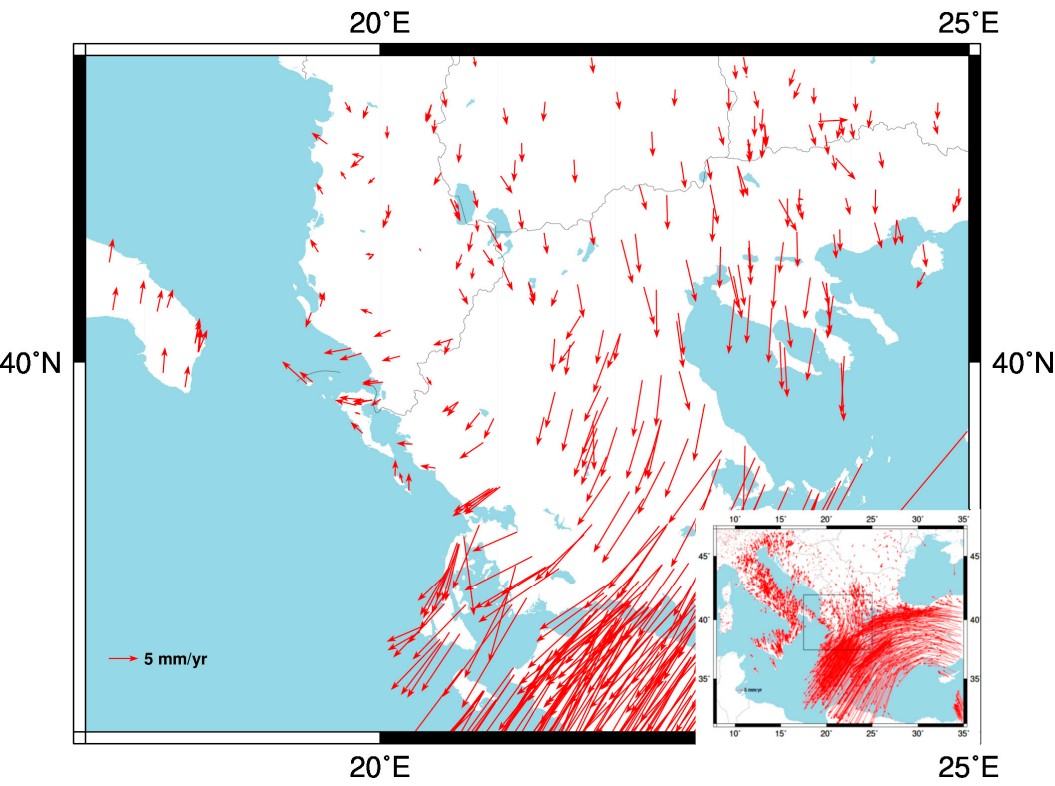

**Figure 2.** Velocities of permanent GNSS Stations in the ETRF2000 "European fixed" reference frame. The velocities on the western side, mostly in Italy, are directed northwards, and those on the central eastern side are directed southwards, implying a dextral shear field. The black rectangle in the insert indicates the area of study. Source: http://pnac.swisstopo.admin.ch/divers/dens_vel/combvel_se_all_cmb_grd_east.jpg. See http://geolabpasaia.org/gnss/agi/maps/EU-DenseVelocities.html#6/42.855/21.955 for an animated map.

## 2. Geodetic Data: InSAR and GNSS

Sentinel-1 data for both ascending (20 November 2019–26 November 2019) and descending (25 November 2019–1 December 2019) passes were analyzed in conjunction with GNSS data from permanent sites. The InSAR data indicate a maximum displacement along the line of sight of approximately 10.5 cm, with an uncertainty that can be conservatively set to 10% or 1 cm (one sigma). The uncertainty is mostly controlled by the tropospheric delay and several other causes [11–13]. The interferograms for the ascending and descending passes are mutually consistent and indicate a very similar deformation pattern. This leads to the conclusion that the atmospheric delay was small enough to maintain correlation between the two interferograms. The direction of flight is ca. 346 degrees for the ascending pass, close to the strike of the fault, and ca. 194 degrees for the descending orbit, with a look angle of ca. 34 degrees of off-nadir angle (Figure 3).

A preliminary calculation based on a model of dislocation in an elastic half space [14,15], with Young's modulus E = 80 GPa and Poisson ratio ν = 0.25, uses the measured (Table 1) moment tensor data to constrain fault geometry (strike and dip of the causative fault) and slip and fault dimensions. The empirical Wells and Coppersmith formulas [16] were used to infer slip and slip area from the momentum magnitude. A model based on a rectangular fault with uniform slip was assumed. This calculation resulted in a maximum surface displacement along the line of sight not larger than 4 cm.

To resolve the discrepancy with the InSAR data, the hypocentral depth of the main event was relocated at a shallower depth. The tests indicated that a depth of 8 ± 2 km was able to result in the observed surface displacement, while keeping the same moment magnitude and orientations of the fault planes (Figure 4). The model uncertainty was estimated as that range of depths such that the resulting line of sight displacements are within 10% of the measured 10.5 cm displacement.

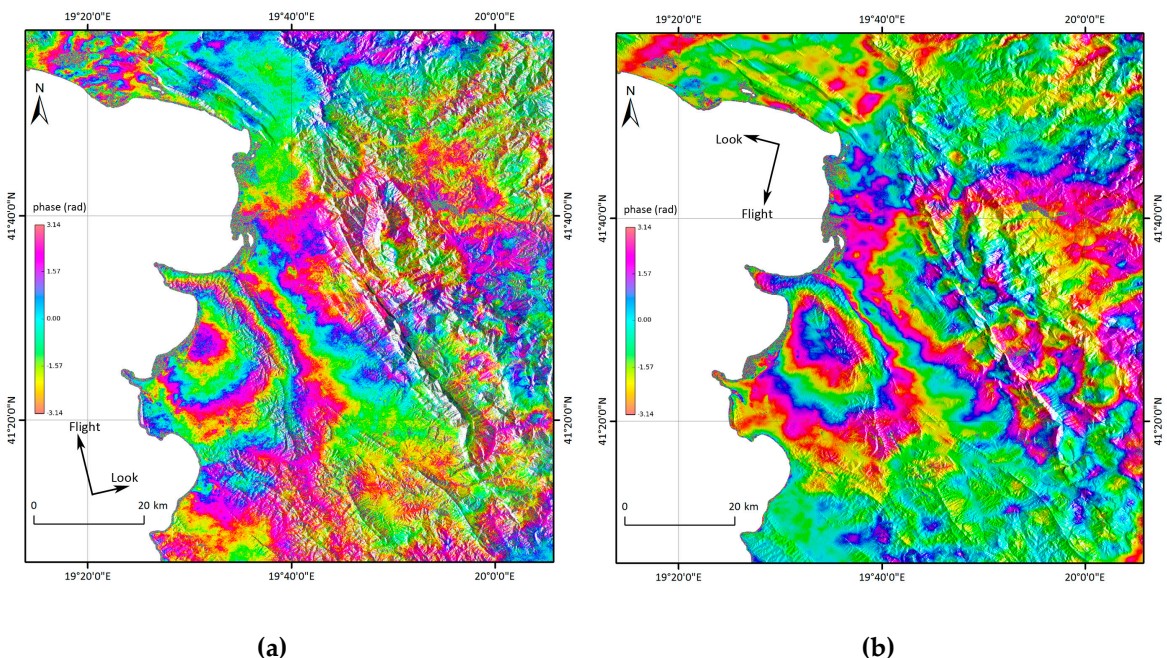

**(a)** **(b)**

**Figure 3.** Interferograms for the (**a**) ascending and (**b**) descending passes of Sentinel 1.

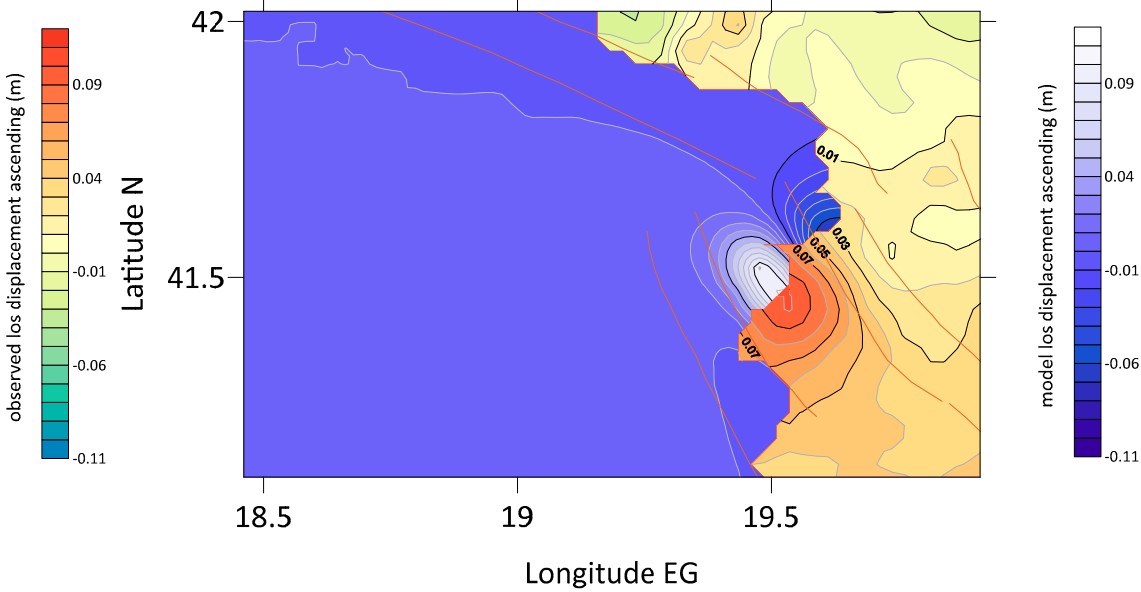

**Figure 4.** Displacement map in the direction of the line of sight (LOS) of the ascending pass of the Sentinel-1 satellite. In the foreground is the observed displacement, truncated at the shoreline, and in the background is the modeled displacement assuming a hypocentral depth of 8 ± 2 km. Brown lines are the fault structures from the project SHARE [1,2]. The plot for the descending pass has negligible differences.

Several permanent GNSS stations belonging to the Albanian permanent networks are routinely processed as part of the Densification of the European Permanent Network [17] and of the Central European GNSS Research Network CEGRN [18–20] using state of the art processing standards [21]. The time series of the horizontal and vertical coordinates available at http://147.162.183.197/ALBANIA/ show sudden discontinuities across the date of the 2019 event with higher amplitudes for the stations (DUR2 and TIR2) near the epicenter, indicating that the stations were displaced by the main event (Figure 5 is an example for DUR2). The previous and subsequent events, being of magnitude 5.5 or less, leave no appreciable signature in the time series, but were nevertheless included in the deformation model. Table 2 gives the coseismic displacements obtained by comparing the horizontal coordinates (vertical coordinates have insufficient accuracy) of the stations before and after the main shock, together with the displacements predicted by the elastic model used for the InSAR data (Table 1, with the hypocentral depth of the main event moved from 26 to 8 km). Table 2 indicates that Durazzo (DUR2) was the station with the highest horizontal displacement (ca. 2 cm). Figure 6 describes the measured horizontal displacements of the processed GNSS sites and the expected vertical motion. The low dip angle plane was assumed as likely principal plane, based on the fact that reverse faults at low dips are activated by a smaller deviatoric stress than at higher dip angle [22]. The green segments in the SW corner indicate the intersection of the assumed fault planes with the Earth surface, at sea. Permanent displacements recorded by a high-resolution digital accelerometer installed at an epicentral distance of ~15 km and probably on the hanging wall of the causative fault are comparable with our GNSS values [23]. Duni and Theodoulidis [23] reported a horizontal Peak Ground Acceleration (PGA) of 1.92 ms$^{-2}$, in good agreement with the average predicted value by a regional model, and a spectral acceleration of at least 5 ms$^{-2}$, for a wide range of periods between 0.3 and 1.0 s.

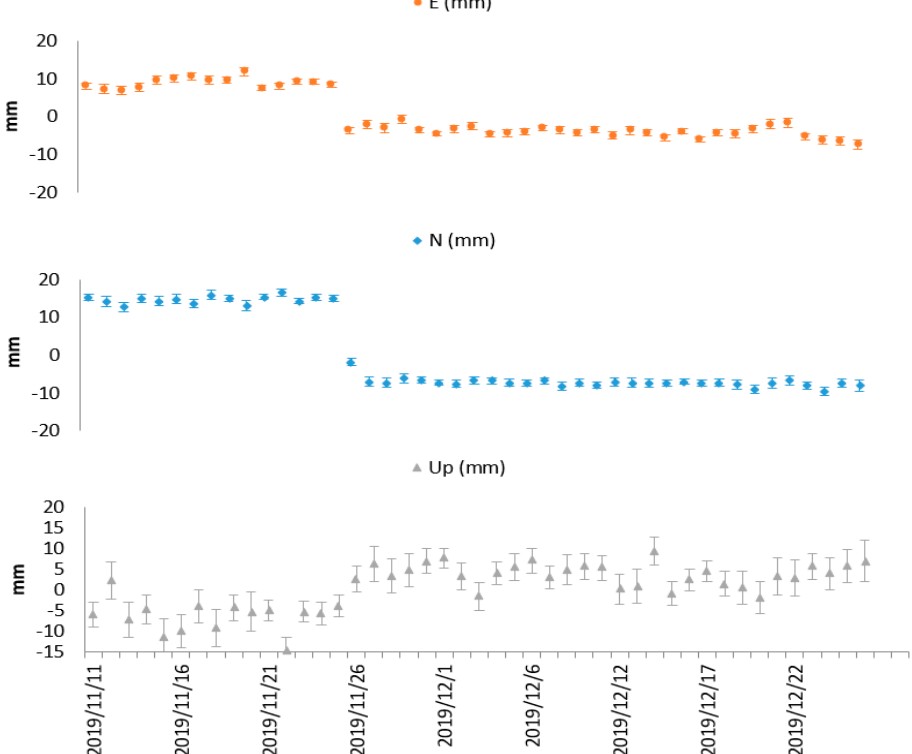

**Figure 5.** Daily time series of the displacement of the GNSS station DUR2 in the east, north, and up directions. The coseismic displacement of 26 November 2019 is in the SW direction and upwards. Error bars correspond to one sigma formal uncertainty.

**Table 2.** Coseismic displacements of selected GNSS stations across the epoch 26 November 2019, 2:54 UTC vs. displacements computed with a dislocation model of an elastic half space based on the moment tensor solution of the main earthquake. The differences between observed and computed displacement in the east and north components have a mean of 0.003 and 0.000 m, respectively, and a root mean square of 0.004 m for both components. Computed displacements are based on Table 1 data with the depth of the main event at 8 km.

| Longitude (deg) | Latitude (deg) | ObsEasting (m) | ObsNorthing (m) | CalcEasting (m) | CalcNorthing (m) | Station Name |
|---|---|---|---|---|---|---|
| 19.945 | 40.708 | 0.001 | −0.002 | 0.000 | 0.000 | BERA |
| 19.451 | 41.316 | −0.012 | −0.018 | -0.018 | −0.025 | DUR2 |
| 19.758 | 40.089 | 0.001 | −0.002 | 0.000 | 0.000 | HIMA |
| 20.698 | 40.707 | 0.001 | −0.001 | 0.000 | 0.000 | KOR2 |
| 20.773 | 40.624 | −0.005 | 0.001 | 0.000 | 0.000 | MALQ |
| 20.440 | 41.685 | −0.002 | −0.003 | −0.004 | −0.002 | PESH |
| 19.875 | 41.768 | −0.003 | −0.003 | −0.009 | −0.010 | RRES |
| 19.496 | 42.051 | 0.002 | −0.003 | 0.000 | −0.001 | SHKO |
| 19.810 | 41.336 | −0.004 | −0.004 | −0.011 | 0.003 | TIR2 |

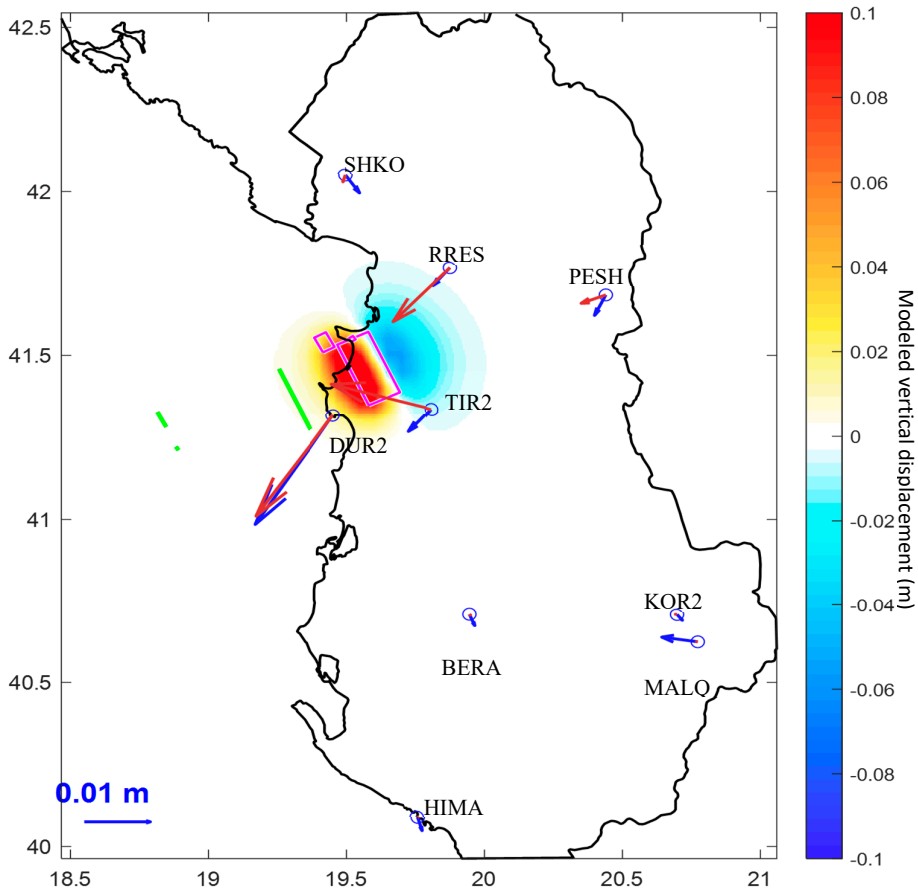

**Figure 6.** Measured (blue arrows) vs. modeled (red arrows) coseismic displacements at permanent GNSS stations in Albania. Purple rectangles represent the fault planes and the green segments the intersections of the fault planes with the Earth surface, under the hypothesis that the low dip faults were activated. The modeled vertical motion is shown with filled contours.

An important question is whether the November 2019 event was in some way affected by the Mw = 7.2 event which took place in Ulcinj (Montenegro) in 1979, ca. 80 km NW of the epicenter of the 2019 event. To this purpose, the change in elastic stress generated by the Montenegro earthquake was projected on planes parallel to the 2019 Albania earthquake using again the deformation of an elastic half space. The Montenegro fault plane was inferred by the pertinent structure MECS001 described

in the DISS (Strike = 300°, dip = 28°, and rake = 90°) at a depth of 8 km. The map shows that the transferred stress at the depth of 8 km, the estimated hypocentral depth of the 2019 Albania earthquake, was in the order of 0.1 bar or less, making it unlikely that the 2019 event was triggered by the 1979 earthquake (Figure 7). A friction coefficient of 0.4 was assumed in the calculation of the Coulomb stress.

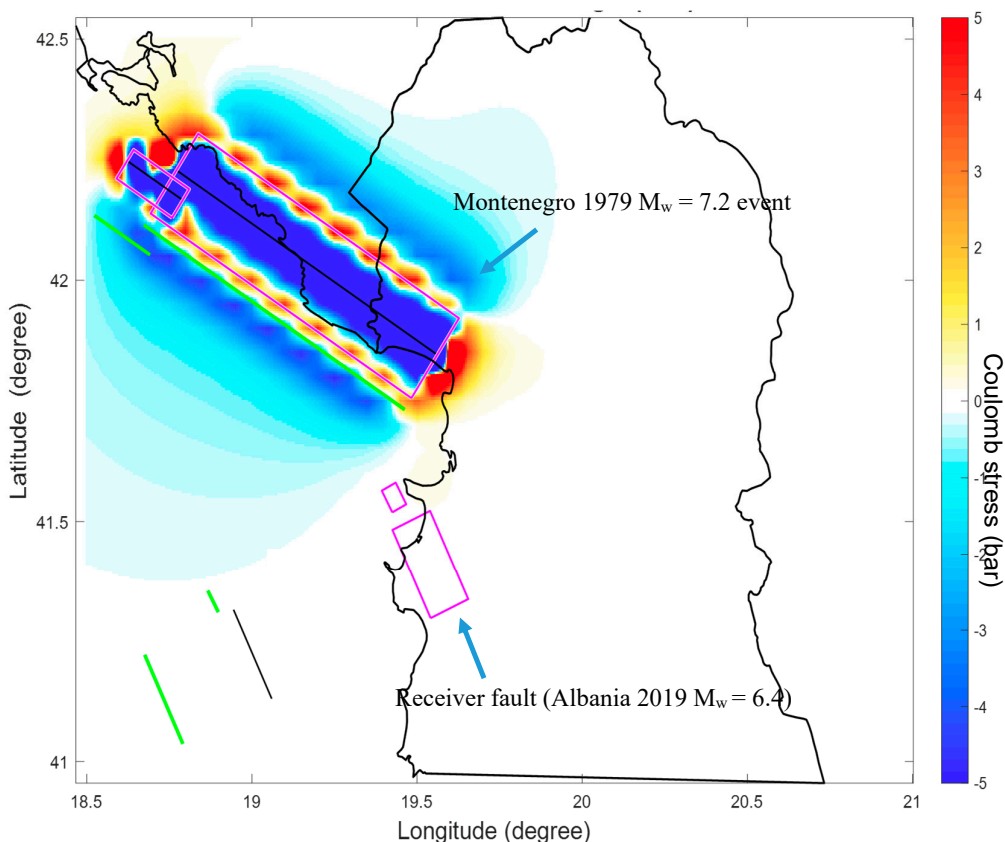

**Figure 7.** Coulomb stress change generated by the 1979 Montenegro event of Mw = 7.2 is projected onto planes parallel to the 2019 Albania earthquake. The amount of transferred elastic stress is negligible. Units: 1 bar = 0.1 MPa.

The maximum stress change resulting from both the 1979 and 2019 events is likely to illuminate a specific area and fault geometry in between the two epicenters. The location and fault angles with the highest Coulomb stress change were therefore investigated, to identify a fault location and geometry with optimal characteristics from the point of view of alignment of the fault to the principal axes of the transferred stress. The analysis indicates that a relatively high shear stress is transferred along a dextral strike in the range 220–240 degrees, dipping nearly 90 degrees and with a rake of ca. 180 degrees. The depth of computation is 8 km, based on the hypocentral depth of the 2019 event. Thus, a likely candidate to accommodate this shear stress is the Lezhe fault (Figure 8). As a consequence of the 1979 and 2019 events, the offshore part of this fault received an extra stress of 2–3 bars, or 0.2–0.3 MPa. This stress change adds to a regional stress of unknown magnitude, but well represented by a plane stress with compressional axis in the same direction SW–NE. If the strain rate is some 30 nstrain/year (1 nstrain = $10^{-9}$) [18], and assuming a shear modulus $\mu$ = 30 GPa, the regional stress rate is on the order of ca. 0.9 kPa/year. Consequently, a seismically induced stress change of 0.2–0.3 MPa (2–3 bar) would imply a time advance of some 2–3 centuries in the stress buildup process at or near the Lezhe fault.

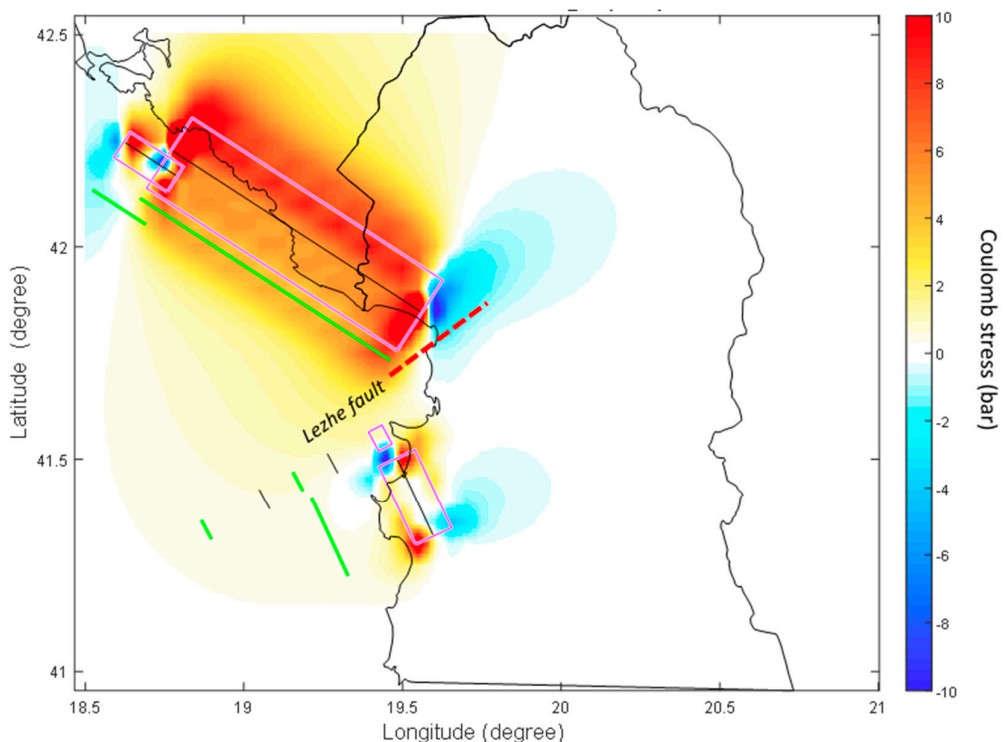

**Figure 8.** Coulomb stress change resulting from the combined action of the 1979 (Montenegro Mw = 7.2) and 2019 (Albania, Mw=6.4) earthquakes, mapped to a plane of strike = 240, dip = 89, and rake = 180 degrees. The Lezhe fault area appears to be positively loaded. The black segments represent the computation depth relative to the mean depth of the fault (purple rectangle). Units 1 bar = 0.1 MPa.

## 3. Conclusions

The November 2019 Albania earthquakes fit very well into the picture of an Adriatic crust subducting under the Albanides/Dinarides. While the epicenters and the fault angles are well constrained, the literature reports a relatively wide range of hypocentral depths, ranging from 9 km (https://earthquakes.ga.gov.au/event/ga2019xgdrrf) to 26 km (http://geofon.gfz-potsdam.de/eqinfo/event.php?id=gfz2019xdig). In most cases, these differences in estimated hypocentral depths reflect the uncertainty in the adopted velocity model. We point out that, based on the InSAR data of Sentinel-1 and on the permanent GNSS stations in Albania, the observed ground displacement requires a shallow hypocenter, of the order of 8 km. We do however expect that a locked zone extends down to some 26 km based on the change of direction of the regional velocity field of GNSS stations and a simple 'arctangent' model [18].

With the improved re-location of the hypocentral depth of the 2019 event, we have further examined the relationship between the 1979 Montenegro earthquake (Mw = 7.2) and the 2019 Albania earthquake (Mw = 6.4) within a model of dislocation in an elastic half space. The model calculations suggest that the 1979 Montenegro event generated a negligible Coulomb stress on the receiver fault likely to have been activated in 2019. Nevertheless, when the two events are considered together again in a purely elastic half space, a non-negligible stress load is visible in the area in-between the two (1979 and 2019) earthquakes. We speculate that the Lezhe fault, a strike slip fault with nearly optimal orientation, could be a likely candidate to receive such an extra load.

**Author Contributions:** M.F. and X.C. analyzed the InSAR data; B.N. provided the GNSS data; M.B. and J.Z. analyzed the GNSS data; and A.C. did the Coulomb stress calculations and wrote the paper. All authors have read and approved the final manuscript.

**Funding:** This research was supported by the Grant "Attività di monitoraggio del territorio veneto a supporto degli strumenti di pianificazione territoriale regionale attraverso il ricorso ai dati forniti dai sistemi GNSS" (Monitoring the territory of the Veneto region for improved mapping using GNSS data) financed by the Regione del Veneto.

**Acknowledgments:** The authors would like to thank Elmar Brockmann of Swisstopo (Wabern, CH) for the dense velocity data set for Europe, and the anonymous reviewers for their constructive and valuable suggestions on the earlier drafts of this manuscript.

**Conflicts of Interest:** The authors declare no conflict of interest.

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
