# Peer review of "The November 2019 Seismic Sequence in Albania: Geodetic Constraints and Fault Interaction"

_remotesensing, doi:10.3390/rs12050846_

Round 1

Reviewer 1 Report

Review of “The November 2019 seismic sequence in Albania: Geodetic constraints and fault interactions”, by A, Caporali et al.

This is an interesting paper, but I feel that there are a lot of details that need to be fleshed out in a revision. For instance, there is an ambiguity between whether you are considering a point fault source (which is in the focal mechanisms) or a finite rupture plane.

How do you compute the surface displacements from the point source focal mechanism (see lines 80-84)? You are using Okada’s solution, which requires you do know finite fault geometry (width and length) and slip (strike slip and dip slip), in addition to depth, dip, and location of the fault plane. The focal mechanism only gives you depth, dip, and location, as well as the rake, but what is the slip magnitude and fault dimensions assumed? 

The surface displacements will vary strongly with the assumed fault dimensions and slip magnitude, and thus I feel that you need a more thorough exploration of this before you can reject the focal mechanism.

There can be strong trade-offs between inferred fault dip and depth in the focal mechanism, and thus if you lower the depth of the fault plane, can you verify that the dip is still the optimal?

In the Coulomb stress transfer models, you definitely do assume finite fault sources. I suspect that the geometry of the 2019 earthquake is what was used when testing the focal mechanism solution, but that needs to be clarified. You also need to flesh out the details of how you determine the finite fault geometry of the 1979 earthquake, and what the uncertainties are.

In lines 145-148 you state that the CSC transfer will advance the stress build-up by 2-3 centuries. This is an interesting result, but missing a lot of details that should be expanded upon. First, can you cite a source for the 0.9 KPa/yr and discuss where that value comes from? How certain is it, and how would the variability of the stressing rate affect your time advance calculation. It is all linear, so there is no inherent reason that you only do the calculation for a single assumed stressing rate. The other issue here is that you need to provide context of the time advance. Is this an advance of time for reloading process on order of hundreds of years, or tens of thousands of years? In the former, the time advance is quite significant, and in the latter it is not. Perhaps reframe this as looking at time advance divided by expected reloading time, in other words for given stressing rates, what is the percentage of expected recurrence time that is advanced.

Author Response

Response to Reviewer 1

Thank you for the very valuable remarks. With their help we have made changes to the text which is now clearer. Here the details:

> there is an ambiguity between whether you are considering a point fault source (which is in the focal mechanisms) or a finite rupture plane.

How do you compute the surface displacements from the point source focal mechanism (see lines 80-84)? You are using Okada’s solution, which requires you do know finite fault geometry (width and length) and slip (strike slip and dip slip), in addition to depth, dip, and location of the fault plane. The focal mechanism only gives you depth, dip, and location, as well as the rake, but what is the slip magnitude and fault dimensions assumed? 

Agree, text has been amended at line 90 and ff. We mention that slip and fault size were estimated from the momentum magnitude using the standard Wells and Coppersmith formulas. A rectangular fault with uniform slip is the basic assumption. Likewise for the 1979 earthquake. The DISS has considerable details on this source and we make reference to them.

>In lines 145-148 you state that the CSC transfer will advance the stress build-up by 2-3 centuries. This is an interesting result, but missing a lot of details that should be expanded upon. First, can you cite a source for the 0.9 KPa/yr and discuss where that value comes from? 

Reference 13 provides a detailed analysis of present day, regional surface deformation rate from geodetic data, including the estimate of a locking depth of about 26 km in the epicentral area obtained by fitting an arctangent model to the GNSS velocities. The profile used for this purpose (see the last figure in ref. [13]) happens to cross exactly the epicentral area. So the used strain rate estimate is tailored on the study area.

>The other issue here is that you need to provide context of the time advance. Is this an advance of time for reloading process on order of hundreds of years, or tens of thousands of years?

We think that so little is known about the Lezhe fault and, more generally, historical seismicity of Albania in the past -say- 1 kyr that a statement on the significance of the time advance more detailed than what we already wrote would be hard to defend. So we prefer to stick to the data we have. My personal guess is that the estimated time advance is significant because the recurrence time for a Mw=6.5 event in a seismic province of some 10000 km2 in that area is likely tobe between 0.5 and 1 kyr, but again a statement which goes beyond a guess should be based on a regional Gutenberg Richter which is not available.

Reviewer 2 Report

All comments and questions are highlighted directly in the text. In general it is a very good article that can be even better after adding some missing images, tables, comparisons, respectively correcting several of them.

Author Response

Response to Reviewer 2.

Thank you for your valuable comments and for the annotated copy. In the following we describe how your remarks and suggestions have been embodied into the revised version of the manuscript.

Specific remarks on the original manuscript

Links to web pages have been double checked.

Page 2: 'Figure 2' was corrected in 'Figure 1'; Figure 1 has been modified to include previous seismicity. Tectonic lineaments were redrawn in yellow for better visibility, as suggested. Scale was added.

Page 3: Figure 2 was redrawn as a detail map plus an insert with a general overview of the velocities, as suggested; links have been checked. More detail on the deformation theory of an elastic half space ('Okada' model) has been added.

Page 4: The original interferograms for both ascending and descending passes have been added, with full information of flight direction and look angle. Reference to the 'green segments' has been removed because inessential. We have not compared in detail GNSS and DINSAR displacements because we feel them complementary: one is horizontal and the other along the line of sight. We think that the figures enable to verify an overall agreement between the two independent data sources. We have added reference to accelerometric data [19] which also provide comparable displacements.  An additional figure exemplifies the coseismic coordinate change at the DUR2 GNSS site. We have corrected the link where additional time series are available.

Figure 4 (Figure 6 in the revised manuscript): scale of the velocities has been changed and all the stations in Table 2 are plotted, as requested.

Reviewer 3 Report

The authors presented interesting research based on the evaluation of seismic displacements using permanent GNSS stations active before and after the seismic sequence that affected Albania. The information and the results achieved provided important and clear details about the earthquake occurred. From a technical point of view, the article is well written and deals effectively and concisely with a very important topic.

Some considerations need to be clarified/integrated. In conclusion, the article can be accepted for publication after the following minor revisions are followed.

----------------------------------------------------------------------------------------------------------------------------------

  1. Query 1: The authors should indicate the accelerograms recorded associated to the referred case study earthquake. In this case, a better interpretation of the earthquake effects released on the case study area are considered;
  2. Query 2 (Figure 1): The authors should geo-reference Figure 1 (and the others);
  3. Query 3 (line 78-86): It is better to indicate how the authors have considered the uncertainty model associated to the maximum displacement achieved by InSAR;
  4. Query 4 (line 103): The period was interrupted by incorrect punctuation. Please recheck;
  5. Query 5: In general, it should be better to associate the occurred PGAs recorded to the displacements achieved. In this case it is possible to have a global overview regarding the case study event. In particular, are the associated PGAs (horizontal and vertical) critical for this fault type? How is the V/H ratio considering the focal depth at 8 km? Is the influence of vertical displacements related to the PGAs occurred? Which is the influence of ground motion vertical component?

Author Response

Response to Reviewer 3

Thank you for your very valuable comments. In the  following we report how they were used to improve the original version of the manuscript.

  • Query 1: The authors should indicate the accelerograms recorded associated to the referred case study earthquake. In this case, a better interpretation of the earthquake effects released on the case study area are considered;

Agree. We have added  a text referring to reference 19 which was devoted to accelerometric data. This makes the analysis more complete. 

  • Query 2 (Figure 1): The authors should geo-reference Figure 1 (and the others);

Agree, we have added a scale and made more precise the insert.

  • Query 3 (line 78-86): It is better to indicate how the authors have considered the uncertainty model associated to the maximum displacement achieved by InSAR;

Agree, we have made reference to specific review papers and explained why our estimate of the uncertainty is likely to be conservative.

  • Query 4 (line 103): The period was interrupted by incorrect punctuation. Please recheck;

Done

  • Query 5: In general, it should be better to associate the occurred PGAs recorded to the displacements achieved. In this case it is possible to have a global overview regarding the case study event. In particular, are the associated PGAs (horizontal and vertical) critical for this fault type? How is the V/H ratio considering the focal depth at 8 km? Is the influence of vertical displacements related to the PGAs occurred? Which is the influence of ground motion vertical component?

We have added a specific reference to a report describing the accelerometric data (Reference 19). Here the authors point out that they expect horizontal displacement at Durazzo of few cm to the SW, which agrees with the GNSS data. They also point out that their work is based on a preliminary velocity model and on a hypocentral depth that is -in our view- too large. So they warn that their results have to be considered as preliminary suggestions.

Reviewer 4 Report

This paper is related to rather geology than remote sensing. Though InSAR and GNSS data are used, the main topic of this paper is inversion analysis of the earthquake and its regional effect. In addition several unclear parts are found.

1. The external links (L. 34, 38 and so on) should be listed as references. Please confirm the validation of them as well.

2. Though the authors cited more realistic epicenter depth (9 km from https://earthquakes.ga.gov.au/event/ga2019xgdrrf) in the conclusion, why the authors cited only 26km one in the other part of this article? Isn’t 26km one a preliminary report?

3. Is the direction of the flight in L. 78-79 north top? Please clarify.

4. In L. 80, "a look angle of ca. 34 degrees relative to nadir" should be "approximately 34 degrees of off-nadir angle". On the other hand, as the Sentinel-1 has wide swath, it should be converted to the local incidence angle.

5. All GNSS stations in Tab. 2 should be shown at least in Fig. 1 or 2.

6. The authors discussed the relationship between 1979 Montenegro earthquake however; it seems too distant in both temporal and spatial aspect. It seems that the authors carried out the discussion only to lead the discussion for Lezhe fault. If the authors want to discuss among them, it is better to do it in more specific journal.

Author Response

  1. The external links (L. 34, 38 and so on) should be listed as references. Please confirm the validation of them as well.

Thank you. We will check with the editorial office

2. Though the authors cited more realistic epicenter depth (9 km from https://earthquakes.ga.gov.au/event/ga2019xgdrrf) in the conclusion, why the authors cited only 26km one in the other part of this article? Isn’t 26km one a preliminary report?

Agree. In the caption of Table 1 we emphasize that the quoted hypocentral depth is debated. In Section 3 we provide a range for the depths available in the literature. Our estimate is on one extreme of the range, so we feel meaningful to compare the InSAR data  with a model based on the other extreme of the range (26 km), to see the difference with the model based on a hypocentral depth of 8 km.

3. Is the direction of the flight in L. 78-79 north top? Please clarify.

Agree. We have added for more clarity the interferograms for the ascending and decending passes with full information.

4. In L. 80, "a look angle of ca. 34 degrees relative to nadir" should be "approximately 34 degrees of off-nadir angle". On the other hand, as the Sentinel-1 has wide swath, it should be converted to the local incidence angle.

Agree. Text has been modified as suggested.

5. All GNSS stations in Tab. 2 should be shown at least in Fig. 1 or 2.

Agree. The relevant figure was modified to include all the stations

6. The authors discussed the relationship between 1979 Montenegro earthquake however; it seems too distant in both temporal and spatial aspect. It seems that the authors carried out the discussion only to lead the discussion for Lezhe fault. If the authors want to discuss among them, it is better to do it in more specific journal.

We agree in part. Crustal rocks can store elastic stress for centuries before releasing it or transforming to permenent deformation. The 1979 event was of quite large magnitude, and we felt that the issue of interaction of the two events should be addressed. The theme of the Special Issue is on deformations measured by InSAR and GNSS: we feel that the geophysical interpretation of these deformations is appropriate in this context.

Round 2

Reviewer 4 Report

The revised manuscript seems fine to be published.